# Family and personal factors predisposing adolescents to substance abuse in high-risk urban areas

María Quinde Reyes[1,2,3], Francisco Jiménez Bautista[2], Nadia Soria-Miranda[1], Daniel Oleas[4], Guido Mascialino[5], Isaac Vera Ponce[3], Jose A. Rodas[6,7]*

1 Facultad de Ciencias Psicológicas, Universidad de Guayaquil, Guayaquil, Guayas, Ecuador, 2 Departamento Paz y Conflicto, Universidad de Granada, Granada, Andalucía, Spain, 3 Centro Psicológico de Especialidades, Guayaquil, Guayas, Ecuador, 4 Escuela de Psicología, Universidad Ecotec, Samborondón, Guayas, Ecuador, 5 Escuela de Psicología y Educación, Universidad de las Américas, Quito, Pichincha, Ecuador, 6 Escuela de Psicología, Universidad Espíritu Santo, Samborondón, Guayas, Ecuador, 7 School of Psychology, University College Dublin, Dublin, Leinster, Ireland

* jose.rodas@ucd.ie

## Abstract

### Objective

This study explored the influence of personal and familial factors on adolescent substance abuse in high-risk urban areas. The aim was to identify psychological and family-based predictors of abuse to inform more comprehensive prevention strategies.

### Method

A comparative cross-sectional design was applied to a sample of 60 adolescents aged 12–18 from vulnerable neighbourhoods in Guayaquil, Ecuador. Participants were divided into two groups: those with substance use disorder (SUD) and those who did not use drugs at all. Emotional stability and cognitive control were assessed using the Cuestionario de Personalidad Situacional, while family dynamics were evaluated with the McMaster Family Assessment Device and the Risk Family Characteristics Inventory. Welch's t-tests and chi-square analyses were conducted for group comparisons, and multivariate logistic regression identified predictors of drug abuse.

### Results

Both groups demonstrated below-average scores in cognitive control and emotional stability, with substance abuse participants showing significantly lower cognitive control. Contrary to expectations, overall family functioning did not significantly differ between groups. However, adolescents with substance abuse were more likely to have family members with substance abuse issues. Logistic regression revealed that

**Data availability statement:** The data underlying the results presented in the study are available from OSF at https://osf.io/5ftuj/?view_only=e2a569f8e49a4e6e837d6357ddaa2e7c.

**Funding:** University College Dublin funded the publication of the present work. The funders had no role in study design, data collection and analysis, decision to publish, or preparation of the manuscript.

**Competing interests:** The authors have declared that no competing interests exist.

critical health problems in the family, poor family problem-solving abilities, and low emotional stability were significant predictors of drug abuse, with respective odds ratios indicating increased risk.

## Conclusion

The findings underscore that adolescent substance abuse is shaped by both individual psychological vulnerabilities and specific family stressors, rather than by global family functioning alone. Effective interventions should integrate support for emotional regulation with family-based strategies that enhance problem-solving and address critical health burdens. These results suggest a need to reframe prevention approaches to target both personal and environmental risk mechanisms simultaneously. While the study's cross-sectional nature and reliance on self-reports limit causal inferences, its findings provide valuable direction for future longitudinal and intervention-based research.

## Introduction

Adolescent drug and alcohol abuse continues to be a significant public health concern globally, profoundly affecting individuals, families, and entire communities. According to the Global Burden of Disease Study 2019, 494,492 people died in 2019 globally due to drug abuse, and 128,083 due to substance use disorder (SUD). In Latin America alone, the number of deaths are 20,375 and 3,019 respectively [1]. The pervasive impact of substance abuse extends beyond mortality, contributing to a wide range of health and social problems, including increased risk for mental health disorders, infectious diseases, and diminished life opportunities.

The burden of these issues is not uniformly distributed, with significant variations observed across different regions and within specific communities. Prior research identified social determinants such as socioeconomic status, access to education, and community resources that play important roles in the prevalence and consequences of adolescent substance abuse [2]. For instance, disadvantaged neighbourhoods often experience higher rates of substance abuse, pointing to the need for targeted interventions that address both individual and environmental risk factors [3].

Substance abuse also impacts adolescents' development in different aspects. For instance, it can significantly hinder physical, psychological, and social development during adolescence, a critical period for cognitive maturation and social identity formation. Research indicates that substance abuse during these formative years can lead to long-term impairments in brain function, potentially resulting in decreased cognitive abilities, altered reward systems, and increased risk of psychiatric disorders [4]. Furthermore, adolescents who engage in drug use are more likely to experience educational disruptions, social isolation, and increased conflict within family and peer relationships, complicating their transition into adulthood [5].

There are several factors involved in substance abuse among adolescents, ranging from personal attributes to familial environments [6,7]. Personal factors such as

emotional stability and cognitive control play important roles in moderating stress responses and impulsive behaviours, which are closely linked to substance use. For instance, adolescents with lower emotional regulation and poor impulse control are found to be more susceptible to substance abuse as they may use them as a coping mechanism [8].

Emotional stability refers to an individual's ability to maintain a balance in their feelings and behaviours, which is crucial during adolescence—a period marked by emotional volatility and developmental changes. This is achieved through the use of emotion regulation strategies leading to wellbeing and adaptive behaviour. The engagement of adaptive emotion regulation strategies leading to emotional stability is a complex process that also involves the use of cognitive resources, particularly those involved in cognitive control [36–38]. Cognitive control involves the ability to regulate and manage one's thoughts and actions, which is critical in resisting impulsive behaviours such as drug use [9,10]. In adolescents, the developmental immaturity of brain regions involved in emotional regulation and cognitive control may contribute to greater vulnerability to substance abuse [11]. Among adults, deficits in these areas have been linked to difficulties in managing drug cravings and maintaining sobriety, suggesting that interventions enhancing emotional regulation and cognitive skills could be beneficial in substance abuse treatment [12].

Research further indicates that the interaction between emotional stability and cognitive control can predict the likelihood of engaging in substance use and the efficacy of substance abuse treatments. For instance, theoretical models suggest that psychosocial stress can initiate physiological arousal and perseverative cognition, which in turn trigger automatic cognitive processes that sustain alcohol use through attentional biases, craving, and maladaptive coping responses [13]. Within this framework, emotional stability may buffer against stress-induced negative affect, while functions related to cognitive control supports the regulation of attentional and decision-making processes that could otherwise perpetuate substance use [39].

Within families, factors such as the presence of substance abuse among family members and the effectiveness of problem-solving strategies are known to either mitigate or exacerbate drug use in adolescents [14]. A family history of substance abuse can predispose adolescents to similar challenges, while strong family bonds and effective communication are protective factors against substance abuse.

The influence of familial factors on adolescent substance abuse extends beyond genetic predispositions [43], encompassing the quality of family relationships and parental practices. Research has consistently shown that adolescents are less likely to engage in substance use when they experience warm and supportive parenting [44], clear boundaries, and consistent discipline within the family [15]. Conversely, conflictual or detached family relationships can increase the risk of substance abuse as adolescents may turn to drugs to cope with stress or emotional pain. The presence of parental substance abuse significantly influences the behaviours and future risk factors for children within those households, fostering a cycle of substance abuse that is difficult to break. When parents use drugs, their children may observe and eventually mimic these behaviours, perceiving them as normal or acceptable ways to cope with stress or emotional distress. This modelling effect is not merely about imitation but also involves the normalisation of substance use as a coping mechanism within the family setting [16].

Furthermore, children in these environments often experience a range of adverse consequences including emotional neglect or abuse, inconsistent parenting, and exposure to the drug-related activities of their parents [17]. These factors collectively contribute to increased vulnerability to substance use among these children as they grow into adolescence and adulthood. The intergenerational transmission of substance abuse is compounded by genetic predispositions and environmental factors, creating a complex web of risk that can perpetuate the cycle of substance abuse across multiple generations [18].

Moreover, the role of family intervention programs in preventing and treating adolescent substance use highlights the potential for modifying family dynamics to reduce risks. Such programs focus on improving communication skills, enhancing parental involvement in children's lives, and equipping families with effective conflict resolution strategies. These interventions can lead to significant reductions in substance use among adolescents by fostering a more supportive and secure family environment

[19]. Although not all studies have found these interventions to be effective for treating additive behaviours in the long term [40], a systematic review found that family-based interventions for substance use not only improved the adolescents' outcomes, but also family cohesion and conflict resolution, two key factors in the development of substance use disorder [41]. This suggests that interventions tailored to address specific family dynamics can be crucial in treating or preventing substance abuse among adolescents, offering a practical approach to mitigating these risks through family-based strategies.

However, prior research often focuses on one set of factors, either personal or family variables, without considering the interactive effects with the other, leaving an important knowledge gap. This oversight can hide the complex interaction between individual vulnerabilities and environmental influences, which both contribute to the risk of adolescent substance use. As such, it is essential for future research to adopt a more integrative approach to fully understand the multifaceted nature of adolescent drug abuse [20].

The current study aims to fill these research gaps by systematically investigating how personal and family factors together influence the likelihood of substance abuse in adolescents. By integrating approaches from developmental psychology and family studies, this research will explore how individual vulnerabilities interact with familial structures to influence substance abuse outcomes.

This approach is significant as it could inform more nuanced prevention and intervention strategies. Understanding the specific pathways through which these factors exert their influence will enable the development of targeted interventions that address both personal vulnerabilities and family dynamics. Consequently, this could lead to more effective prevention strategies that are tailored to the nuanced needs of at-risk adolescents, ultimately reducing the incidence and impact of adolescent substance abuse.

## Methods

### Participants

The study sample includes two groups of adolescents, aged 12–18 years. The first group consisted of 30 male adolescents without a history of substance use, with a mean age of 15.43 years (SD = 2.11). The second group comprised 30 male adolescents with substance abuse issues, with a mean age of 15.60 years (SD = 1.71). Adolescents without substance use were users of medical services from three health centres in Guayaquil, located within areas identified as high-risk for drug consumption: Bastión Popular, Vergeles, and Monte Sinaí.

The group of adolescents with substance use issues was recruited from two specialised centres for the treatment of problematic alcohol and drug use (CETAD), namely Los Libertadores and Juan Pablo II. These centres also serve adolescents from the same geographical areas as those without substance use issues. Admission to these centres was for treatment purposes. Ethical approval and access to the participants from both groups were obtained from the District Directorate of the Ministry of Public Health of Ecuador, with additional consent obtained from the parents or guardians.

In terms of family background, half of the adolescents in the total sample originated from nuclear families. Educational attainment among parents varied between the groups: parents of the adolescents without substance use most frequently had university-level education, whereas the parents of adolescents with substance use predominantly had only completed primary and secondary education. Educational environments also differed; adolescents without substance use attended private educational institutions, whereas those with substance use, who were institutionalised at CETADs due to habitual substance use, often had histories of legal infractions, including contract killing and theft.

### Instruments

**Personality.** To assess personal factors, the Cuestionario de Personalidad Situacional (Situational Personality Questionnaire; CPS) was employed, developed on the recognition of stable personality traits and the behavioural dimensions pertinent to various situational and contextual settings [21]. This instrument, originally developed in Spanish, is accompanied

by norms applicable to the Latin American demographic and comprises 233 items framed in a true/false format, evaluating 15 different personality traits. For the purposes of this study, only the scales related to emotional stability and cognitive control were used. The scale of emotional stability measures the capacity to regulate and control emotions effectively, including items such as "I become angered by many things," "When angered, I tend to speak loudly," "I experience significant mood fluctuations," "I am easily upset," or "At times, I express my emotions explosively." The cognitive control scale is designed to assess the ability to self-regulate behaviour and manage activities in daily life, including items such as "Many of the things that happen to me are because of bad luck," "I have tact and diplomacy when saying things," "I am careful when organising my work," "I carefully plan activities," or "I keep control over my words during discussions." The total score is calculated by TEA Corrige, an online platform, after introducing all the participant's responses to the system. Although the profile generated from TEA Corrige provides T scores, we used the raw scores for analyses.

**Family risk factors.** Family factors were assessed using two instruments. The first is the Spanish version of the McMaster Family Assessment Device (FAD), which consists of 60 items with four response options ranging from 'strongly agree' to 'strongly disagree'. Items are scored from 1 to 4. The original version was developed by Epstein et al. [22] to evaluate the six dimensions of the MacMaster model of family functioning. This instrument has shown to be useful at both clinical and research settings [23], and has been translated into various languages for use in different contexts with healthy, clinical, and psychosocially troubled populations; it has been used in comparative group studies. In the Spanish adaptation study of the instrument, factor loadings were reported for three dimensions: affective response, family problem solving, and family emotional involvement [24]. For the current study, we used the scales for family problem solving and family emotional involvement, both obtained from the factor analysis reported by Barroilhet at al. [24].

The second instrument used was the Risk Family Characteristics Inventory, designed and validated by Louro [25] for health assessments of family groups in primary care settings. This questionnaire identifies situations that trigger family crises and hinder the development and well-being of family members. It includes 50 items formatted as a checklist (present or not present), with risk characteristics grouped into seven areas: socioeconomic and cultural context of family life, household composition, normative critical processes, non-normative critical processes, health, family coping, and social support. For this study, we utilised the sub-scale of normative critical processes, which assesses the presence of any of the following issues among family members: chronic diseases, poor dietary practices, teenage pregnancy, drug or alcohol abuse, HIV/AIDS, cancer, disabilities, terminal illness, suicidal behaviours, genetic problems, infertility, and abuse, neglect, mistreatment, or abandonment. The analysis was based on the number of problems reported.

## Procedure

The assessment was conducted using paper and pencil at each participating centre. To meet ethical standards, information about the study and written informed consent forms were shared with the participants and family representatives or professional guardians. Only those participants who were willing to participate and returned the consents signed by their legal guardians were included in the study. To minimise application biases, assessments were carried out in air-conditioned spaces isolated from noise and distractions. Participants were evaluated over two sessions, each approximately two hours long. Demographic data for the adolescents and their families were collected. In the first session, tests relating to personal factors were administered, and in the second, those relating to family factors were conducted. Additional questionnaires were also part of the evaluation; however, their results are not reported in the present study. Participants were not provided with financial compensation.

## Ethics

This study was reviewed and approved by the Ministry of Public Health of Ecuador, specifically the District Directorate 09D08 – Pascuales 2 – Salud. The approval was granted under Memorandum No. MSP-CZ8S-DD09D08-DIR-2022–3425-M, dated 29 April 2022. All procedures were conducted in accordance with relevant ethical guidelines and

regulations. Verbal and written informed consent was obtained from the participants and parents or legal guardians of all participating adolescents prior to data collection. Confidentiality and anonymity of participants were maintained throughout the study, and no identifying information was disclosed. Participants and their guardians agreed on their de-identified data to be used for research purposes. Data collection took place between 27 July 2022 and 24 February 2023.

## Analysis plan

The analysis plan for the study was designed to investigate the relationships between various personal and family factors and the presence of substance use in adolescents. Descriptive statistics, including means and standard deviations for continuous variables, and frequencies for categorical variables, were first calculated to provide an overview of the data collected.

For comparative analysis between the two groups of adolescents (those with and those without substance use), the Welch t-test was employed to assess differences in age, emotional stability, cognitive control and family functioning. The chi-square test was used to explore associations between substance use and categorical variables such as family members with substance use problems, suicidal attempts in family members, infertility problems, and experiences of abuse, neglect, mistreatment, or abandonment. To control for the inflation of Type I error due to multiple comparisons, an adjusted alpha level was applied following the Bonferroni procedure.

Furthermore, a multivariate logistic regression model was implemented to identify significant predictors of substance use among adolescents. The model included as predictors several internal and family-related factors: emotional stability, cognitive control, family problem solving, family emotional involvement, and critical health issues within the family. A stepwise selection procedure was employed to refine the model and identify the most influential predictors. Odds ratios and 95% confidence intervals were calculated providing quantified measures of risk. All analyses were conducted using R [35].

## Results

Table 1 summarises the descriptive statistics for the investigated variables, specifically the frequency of family problems reported per group and the mean and standard deviation of other variables. When comparing the groups, no significant differences were found between participants with and without substance use in terms of age, as determined by a Welch t-test (t(55.53) =.269, p =.789). However, a significant association was found between group affiliation (with or without substance use) and having family members with substance use problems ($\chi^2$(1, N=60) = 10.76, p =.001), with the clinical sample more likely to have a family member with substance use problems. Other potential associations were observed, but these did not survive an adjusted alpha for multiple comparisons, such as the presence of suicidal attempts in family members ($\chi^2$(1, N=60) = 4.81, p =.028), infertility problems ($\chi^2$(1, N=60) = 4.04, p =.044), and manifestations of abuse, neglect, mistreatment, or abandonment ($\chi^2$(1, N=60) = 4.32, p =.038). In all instances, participants with substance use had a higher incidence of these issues. Although families in both groups reported critical health issues, these were consistently more frequent among those with adolescents experiencing substance use.

The CPS provides a profile for each participant, detailing both raw scores and normalized S scores. The S scores are standardized with a mean of 50, and a standard deviation represented by 20 points. For adolescents with substance use, the mean S scores are notably low, recorded at 14 (SD = 10.4) for emotional stability and 15.85 (SD = 14.41) for cognitive control, indicating scores considerably below the normative sample. Similarly, the adolescents without substance use also exhibit low scores relative to the normative sample: emotional stability is at 24.17 (SD = 18.57) and cognitive control at 22.43 (SD = 14). A significant difference between the two groups was observed in cognitive control, as indicated by Welch's t-test results (t (51.24) = 2.15, p =.036, d =.57). However, no significant difference was found in emotional stability (t (49.33) = 1.85, p =.071, d =.48). This data suggests that both groups, regardless of substance use status, score low compared to typical developmental expectations.

**Table 1. Descriptive statistics from personal and family factors.**

| Number of cases | No substance use | With substance use |
|---|---|---|
| Chronic disease | 4 | 10 |
| Poor dietary practices | 4 | 6 |
| Teenage pregnancy | 2 | 6 |
| Alcohol abuse | 12 | 13 |
| Drug use | 2 | 13 |
| HIV/AIDS | 3 | 4 |
| Cancer | 4 | 6 |
| Disability | 5 | 8 |
| Terminal illness | 3 | 9 |
| Suicidal behaviours | 3 | 10 |
| Genetic problems | 4 | 7 |
| Infertility | 1 | 6 |
| Abuse, neglect, mistreatment, or abandonment | 2 | 8 |
| **Mean scores (standard deviation)** | | |
| Critical health issues in the family | 1.63 (2.83) | 3.53 (3.4) |
| Family problem solving | 20.2 (5.55) | 17.93 (5.07) |
| Family emotional implication | 30.27 (4.4) | 31.73 (5.91) |
| Emotional stability | 13.43 (6.11) | 10.96 (3.83) |
| Cognitive control | 16.73 (2.59) | 15.11 (3.06) |

The comparison of family functioning between the two groups in our study, using the McMaster Family Assessment Device, revealed statistically similar dynamics across several family functioning domains. Specifically, Welch's t-tests demonstrated no significant difference between the groups in Problem Solving (t(53.242) = 0.811, p = .421), Communication (t(57.144) = 0.035, p = .972), and Affective Responsiveness (t(55.703) = 0.137, p = .891), indicating a similar level of functioning in these areas. Similarly, no significant differences were noted in Affective Involvement (t(57.768) = −0.949, p = .346) and Behavioural Control (t(57.651) = 0.032, p = .974). The Roles subscale did show a statistically significant difference (t(56.984) = −2.124, p = .038); however, this p-value does not survive an adjusted alpha for multiple comparisons. Additionally, General Functioning (t(54.501) = −0.107, p = .915) did not differ significantly, further affirming the comparability of the two groups in overall family functioning. These results collectively suggest that variations in substance use status within our study population are unlikely to be explained by differences in family functioning as measured by the McMaster Family Assessment Device.

The multivariate logistic regression model assessed the likelihood of substance use (present or absent) as the outcome, using both family and internal variables as predictors: emotional stability, cognitive control, family problem-solving, family emotional involvement, and critical health issues within the family. A stepwise selection process was used to identify significant associations between substance use and these predictors. According to our results (refer to Table 2), three factors notably increased the risk of adolescents developing a substance use issue, two of which are family-related. Adolescents encountering critical health issues within their family faced a 29.8% (95% confidence interval [5.8%, 59.3%]) increased risk of developing a substance use issue when controlling for other variables. Similarly, a lack of family problem-solving skills was associated with a 15.5% (95% confidence interval [26.6%, 2.8%]) increased risk, and insufficient emotional stability was linked to a 16.4% (95% confidence interval [27.1%, 4.1%]) higher risk of substance use.

**Table 2. Resulting model from the multivariate logistic regression analysis.**

| Parameter | Estimate | Standard Error | Odds Ratio | Wald Test | | 95% Confidence interval (odds ratio scale) | |
|---|---|---|---|---|---|---|---|
| | | | | Wald Statistic | p | Lower bound | Upper bound |
| (Intercept) | 4.592 | 1.837 | 98.703 | 6.250 | 0.012 | 2.696 | 3613.057 |
| Critical health issues in the family | 0.261 | 0.104 | 1.298 | 6.252 | 0.012 | 1.058 | 1.593 |
| Family problem solving | −0.169 | 0.072 | 0.845 | 5.573 | 0.018 | 0.734 | 0.972 |
| Emotional stability | −0.179 | 0.070 | 0.836 | 6.529 | 0.011 | 0.729 | 0.959 |

*Note.* adiction level '1' coded as class 1.

## Discussion

Our study provides a view of the factors associated with substance use among adolescents. The descriptive statistics detailed the frequency and types of family problems across groups, revealing no significant age and family functioning differences between adolescents with and without substance use. However, a significant association was found between substance use status and having family members also suffering from substance use, suggesting familial patterns in substance use. Furthermore, our findings highlight critical disparities in cognitive control between the two groups, with both scoring below developmental norms but adolescents with substance use demonstrating particularly impaired scores. While other potential associations related to family issues such as suicidal attempts, infertility, and various forms of maltreatment were observed, these did not remain significant after adjusting for multiple comparisons. The multivariate analysis highlights that critical health issues within the family, deficient family problem-solving skills, and inadequate emotional stability significantly increase the likelihood of adolescent substance use, thereby emphasising the importance of both internal and family factors in mitigating such risks. These results underscore the complex interplay between familial dynamics and individual psychological factors in the risk of substance use.

The escalating challenge of adolescent drug and alcohol use underscores the profound impact on public health, individual development, and family dynamics globally [5]. In this context, our study contributes significant insights into the interplay between familial and personal factors influencing substance use in adolescents. Despite the broader social determinants like socioeconomic status and community resources, which have been linked to substance use prevalence [3], our findings highlight the critical roles of family problem-solving skills, emotional stability, and the presence of family members with substance use issues [26]. These factors have emerged as substantial predictors of adolescent substance use, aligning with research that underscores the importance of family dynamics in the development of substance use patterns among adolescents [8,15].

Our data show that adolescents facing critical health issues within their families and those with insufficient family problem-solving skills are more prone to substance use. This is in agreement with studies that highlight the role of familial adversity in increasing substance use risks [27,28]. Moreover, our findings underline the considerable deficits in cognitive control and emotional stability among adolescents with substance use, indicating that these traits significantly contribute to substance use vulnerability [12,29]. These observations are in line with the body of research asserting that impairment in emotional regulation and impulse control during the critical developmental phase of adolescence can lead to enduring cognitive and emotional disturbances, thereby heightening the likelihood of substance abuse [30].

The observed significant association between substance use in adolescents and the presence of family members who also have substance use issues underscores the intertwined individual and environmental factors influencing substance use behaviours, including family related factors. This pattern of familial substance use and within family availability [42] suggests a cycle where the normalisation of substance use in the family can lead to the establishment and perpetuation of an addiction [31,32]. These findings emphasise the need for family programs that not only help individuals deal with their

problems but also aim to change the family behaviours that contribute to substance use issues [33,34]. These interventions could be pivotal in disrupting the intergenerational transmission of substance use behaviours.

It is noteworthy that both samples were very similar across several critical dimensions. Importantly, there were no significant age differences between the groups, suggesting that any observed differences in substance use behaviours are not attributable to age disparities. Both groups also originated from similar high-risk neighbourhoods, characterised by elevated levels of delinquency, low income and drug trafficking, which provides a common environmental context for examining the influence of other variables on substance use. Additionally, family functioning, as assessed by the McMaster Family Assessment Device, indicated comparable levels of family dynamics across both samples. This uniformity in family functioning across the groups allows for a more focused investigation of how other specific familial factors, such as the presence of substance use within the family or family problem-solving skills, impact adolescent behaviour. Thus, the equivalence in these background factors strengthens the validity of our findings and supports the conclusion that the differences observed in substance use can be more directly associated with the unique psychological and familial variables explored in our study. However, educational background could not be systematically collected across participants, and thus potential group differences in this domain could not be examined.

Therefore, the results of our study not only reaffirm the importance of addressing individual psychological factors such as emotional stability and cognitive control in the prevention and treatment of substance abuse, but also highlight the need to consider the quality of the family environment. Interventions that enhance family problem-solving abilities, improve emotional communication, and address the broader family health issues could be critical in reducing adolescent substance use rates. This approach, integrating both personal vulnerabilities and family dynamics, could lead to more effective prevention strategies tailored to the specific needs of at-risk adolescents, thereby reducing the widespread impact of this critical public health issue.

## Limitations

This study presents with several limitations. Reliance on self-report measures, while valuable for accessing personal and subjective experiences, are susceptible to various biases such as social desirability, recall bias, and response bias. Participants may under-report behaviours perceived as socially undesirable, such as substance use, or they may not accurately remember or report family dynamics. This could potentially skew the results and limit the reliability of the associations found. Additionally, self-reporting does not capture the complexity of behaviours and environments that a more objective or multi-method approach might reveal. The study's cross-sectional nature also limits the ability to infer causality from the associations observed; longitudinal studies would be better suited to assess the directions and dynamics of these relationships over time. Furthermore, the study's findings are drawn from a specific high-risk neighbourhood, which may limit the generalisability of the results to other populations or settings with different socio-economic conditions or cultural contexts. Therefore, while the study provides important insights into the factors associated with adolescent substance use, these limitations should be carefully considered when applying the findings to broader contexts.

## Author contributions

**Conceptualization:** María Quinde Reyes, Francisco Jiménez Bautista, Jose A Rodas.

**Data curation:** María Quinde Reyes, Nadia Soria-Miranda.

**Formal analysis:** Nadia Soria-Miranda, Jose A Rodas.

**Investigation:** María Quinde Reyes, Francisco Jiménez Bautista, Isaac Vera Ponce.

**Methodology:** Francisco Jiménez Bautista, Nadia Soria-Miranda, Jose A Rodas.

**Resources:** Isaac Vera Ponce.

**Supervision:** Francisco Jiménez Bautista, Jose A Rodas.

**Writing – original draft:** María Quinde Reyes, Nadia Soria-Miranda, Daniel Oleas, Guido Mascialino, Isaac Vera Ponce, Jose A Rodas.

**Writing – review & editing:** Daniel Oleas, Guido Mascialino, Jose A Rodas.

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
