## [Decision Letter · Decision Letter 0]

15 Aug 2025

PONE-D-25-35455Family and personal factors predisposing adolescents to substance abuse in high-risk urban areasPLOS ONE

Dear Dr. Rodas,

Thank you for submitting your manuscript to PLOS ONE. After careful consideration, we feel that it has merit but does not fully meet PLOS ONE’s publication criteria as it currently stands. Therefore, we invite you to submit a revised version of the manuscript that addresses the points raised during the review process. Please ensure that your decision is justified on PLOS ONE’s publication criteria  and not, for example, on novelty or perceived impact.

We look forward to receiving your revised manuscript.

Kind regards,

Rosemary Bassey, Ph.D.

Academic Editor

PLOS ONE

Additional Editor Comments (if provided):

Reviewers' comments:

Reviewer's Responses to Questions

**Comments to the Author**

1. Is the manuscript technically sound, and do the data support the conclusions?

Reviewer #1: Yes

Reviewer #2: Yes

2. Has the statistical analysis been performed appropriately and rigorously? 

Reviewer #1: Yes

Reviewer #2: Yes

3. Have the authors made all data underlying the findings in their manuscript fully available?

Reviewer #1: Yes

Reviewer #2: Yes

4. Is the manuscript presented in an intelligible fashion and written in standard English?

Reviewer #1: Yes

Reviewer #2: Yes

5. Review Comments to the Author

Reviewer #1: The phrase ‘substance abuse’ is utilised throughout the article, however this phrase is associated with stigma and negatively reinforces substance use and associated discrimination. It would be better to utilise ‘substance use’ throughout the article, even in the case of illicit drug use. Line 167 also states ‘substance users’, which again attaches stigma and moves away from person-centred language. Rephrasing to ‘adolescents who use drugs’ will help to eliminate potential stigma. The following NIH resource outlines appropriate terminology to reduce stigma in research: https://nida.nih.gov/research-topics/addiction-science/words-matter-preferred-language-talking-about-addiction

In terms of consent, was consent obtained from the children themselves? The emphasis is on the signature of the parents/legal guardians, but did the children also provide consent, particularly as the age range was 12-18 years old. If only parental/guardian consent was obtained, did the children feel compelled to participate, even if they did not wish to? This needs to be made clearer within the methods as the focus seems to be on parental/guardian consent.

For the statistical tests, was software utilised? If so, it should be detailed within the methods section.

The results do not mention the gender of participants. Unsure if this has been omitted for a specific reason, such as protecting participants anonymity? This needs to be clarified either in the results section or within the limitations.

Lines 357-359: this sentence links the study results to ‘genetic and environmental factors’ which lead to substance use. However, the study does not explore any genetic factors, merely the relationship between familial substance use and adolescent substance use. I would redraft this sentence to reflect the fact that the study does not explore genetic implications.

The introduction and discussion alike rely on some dated studies, much of which is over 10 years old. Both sections would benefit from the inclusion of some more recent research.

Reviewer #2: Line 83 – 92 - A more detailed definition of both emotional stability and cognitive control (executive function) and the interaction between these two systems would be useful in the introduction, For example the author states that emotional regulation plays a role in controlling behaviours, while cognitive function is defined as controlling actions – it is not clear how you are differentiating between behaviours and actions. Using widely accepted definitions around these elements and their interaction would solve this.

Similarly in lines 93 to 97 you state the interaction between emotional stability and cognitive control can influence substance abuse, but your example only considers the impact of emotional stability and not the role of cognitive function or the interaction. Expanding on this slightly would greatly improve the paper.

Line 123 – 131 is there any evidence around using these interventions with families where parents are using substances specifically? Some consideration of this should be included either here or in the discussion where recommendations are made.

Line 357 – 365 – some consideration /acknowledgement of the additional factor of availability of substances within the home meaning young people have increased opportunity to experiment would be beneficial here.

Line 368 – 379 – disparities in educational background should potentially be mentioned here. Particularly as there are strong known interactions between education and emotional and cognitive control.

6. PLOS authors have the option to publish the peer review history of their article (what does this mean? ). If published, this will include your full peer review and any attached files.

**Do you want your identity to be public for this peer review?** For information about this choice, including consent withdrawal, please see our Privacy Policy .

Reviewer #1: No

Reviewer #2: No

---

## [Author Response · Author response to Decision Letter 1]

4 Sep 2025

Response to reviewers

We would like to thank the reviewers for their insightfull comments.

Reviewer #1

The phrase ‘substance abuse’ is utilised throughout the article, however this phrase is associated with stigma and negatively reinforces substance use and associated discrimination. It would be better to utilise ‘substance use’ throughout the article, even in the case of illicit drug use. Line 167 also states ‘substance users’, which again attaches stigma and moves away from person-centred language. Rephrasing to ‘adolescents who use drugs’ will help to eliminate potential stigma. The following NIH resource outlines appropriate terminology to reduce stigma in research: https://nida.nih.gov/research-topics/addiction-science/words-matter-preferred-language-talking-about-addiction

Substance use in this population is quite common, so we wanted to highlight that the studied population not only used but had issues related to their use. We have modified the wording to avoid stigmatisation and we now use the expresion substance use.

In terms of consent, was consent obtained from the children themselves? The emphasis is on the signature of the parents/legal guardians, but did the children also provide consent, particularly as the age range was 12-18 years old. If only parental/guardian consent was obtained, did the children feel compelled to participate, even if they did not wish to? This needs to be made clearer within the methods as the focus seems to be on parental/guardian consent.

Thanks for noting this. All participation was voluntary. We have updated the methods section to make it clear:

Only those participants who were willing to participate and returned the consents signed by their legal guardians were included in the study.

For the statistical tests, was software utilised? If so, it should be detailed within the methods section.

All analyses were conducted using R. We have updated the Analysis Plan sub-section to include this.

The results do not mention the gender of participants. Unsure if this has been omitted for a specific reason, such as protecting participants anonymity? This needs to be clarified either in the results section or within the limitations.

Thanks for noting this! We forgot to mention that all participants were male. We have updated the Participants sub-section from the methods section to clarify this. The fact that all substance users were male was unintentional. Participants from the comparative groups were chosen to match those from the substance use group.

Lines 357-359: this sentence links the study results to ‘genetic and environmental factors’ which lead to substance use. However, the study does not explore any genetic factors, merely the relationship between familial substance use and adolescent substance use. I would redraft this sentence to reflect the fact that the study does not explore genetic implications.

We agree. Thus, the sentence was rephrased to focus on the investigated variables.

The introduction and discussion alike rely on some dated studies, much of which is over 10 years old. Both sections would benefit from the inclusion of some more recent research.

Thank you for this observation. We agree that this represents a limitation of our manuscript and have therefore revised the introduction, incorporating nine recent studies published within the last decade

Reviewer #2

Line 83 – 92 - A more detailed definition of both emotional stability and cognitive control (executive function) and the interaction between these two systems would be useful in the introduction, For example the author states that emotional regulation plays a role in controlling behaviours, while cognitive function is defined as controlling actions – it is not clear how you are differentiating between behaviours and actions. Using widely accepted definitions around these elements and their interaction would solve this.

Thank you for noting this. We have revised the paragraph to clarify the relationship, explaining how both constructs interact through the use of emotion regulation strategies. We have also added relevant references on this subject. The revised paragraph now reads:

Emotional stability refers to an individual's ability to maintain a balance in their feelings and behaviours, which is crucial during adolescence—a period marked by emotional volatility and developmental changes. This is achieved through the use of emotion regulation strategies leading to wellbeing and adaptive behaviour. The engagement of adaptive emotion regulation strategies leading to emotional stability is a complex process that also involves the use of cognitive resources, particularly those involved in cognitive control [36,37,38]. Cognitive control involves the ability to regulate and manage one's thoughts and actions, which is critical in resisting impulsive behaviours such as drug use [9,10].

Similarly in lines 93 to 97 you state the interaction between emotional stability and cognitive control can influence substance abuse, but your example only considers the impact of emotional stability and not the role of cognitive function or the interaction. Expanding on this slightly would greatly improve the paper.

We have updated the paragraph to better explain the results from the cited study and included more evidence on the role of both processes in the development of substance use.

Line 123 – 131 is there any evidence around using these interventions with families where parents are using substances specifically? Some consideration of this should be included either here or in the discussion where recommendations are made.

We have expanded the paragraph considering current evidence on the effectiveness of such interventions.

Line 357 – 365 – some consideration /acknowledgement of the additional factor of availability of substances within the home meaning young people have increased opportunity to experiment would be beneficial here.

Thanks for this suggestion. We have acknowledged the increased risk of family members with substance use disorder in the development of related issues in adolescents and provided a relevant reference.

Line 368 – 379 – disparities in educational background should potentially be mentioned here. Particularly as there are strong known interactions between education and emotional and cognitive control.

We appreciate this observation; unfortunately, educational background data were not available for all participants, which prevented meaningful group comparisons. We added informationin the paragraph to acknowledge this limitation.

---

## [Decision Letter · Decision Letter 1]

22 Oct 2025

Family and personal factors predisposing adolescents to substance abuse in high-risk urban areas

PONE-D-25-35455R1

Dear Dr. Jose Rodas,

We’re pleased to inform you that your manuscript has been judged scientifically suitable for publication and will be formally accepted for publication once it meets all outstanding technical requirements.

There are however a few minor points to address in finalizing this manuscript for publication. In the abstract and early paragraphs of the introduction the terms drug abuse, substance abuse, and SUD are all used and it is not quite clear if you are making clear differences between these three things or using them interchangeably. Between lines 68 - 76 and 76 - 90 there is some lack of flow in the writing that makes it difficult to follow. I acknowledge that this is in part from adding in new information in response to reviewer requests but feel the narrative could flow more smoothly with some minor tweaks to the structuring of these paragraphs. 

Kind regards,

Rosemary Bassey, Ph.D.

Academic Editor

PLOS ONE

Additional Editor Comments (optional):

Reviewer #1:

Reviewer #2:

Reviewers' comments:

Reviewer's Responses to Questions

**Comments to the Author**

1. If the authors have adequately addressed your comments raised in a previous round of review and you feel that this manuscript is now acceptable for publication, you may indicate that here to bypass the “Comments to the Author” section, enter your conflict of interest statement in the “Confidential to Editor” section, and submit your "Accept" recommendation.

Reviewer #1: All comments have been addressed

Reviewer #2: (No Response)

Reviewer #3: All comments have been addressed

2. Is the manuscript technically sound, and do the data support the conclusions?

Reviewer #1: Yes

Reviewer #2: Yes

Reviewer #3: Yes

3. Has the statistical analysis been performed appropriately and rigorously? 

Reviewer #1: Yes

Reviewer #2: Yes

Reviewer #3: Yes

4. Have the authors made all data underlying the findings in their manuscript fully available?

Reviewer #1: Yes

Reviewer #2: Yes

Reviewer #3: Yes

5. Is the manuscript presented in an intelligible fashion and written in standard English?

Reviewer #1: Yes

Reviewer #2: Yes

Reviewer #3: Yes

6. Review Comments to the Author

Reviewer #1: (No Response)

Reviewer #2: Thank you for your attention to the points raised in the reviews and for amending where required or further explaining your decision making. Rereading the article I feel there a few minor points to address prior to publication. In the abstract and early paragraphs of the introduction the terms drug abuse, substance abuse, and SUD are all used and it is not quite clear if you are making clear differences between these three things or using them interchangeably. Between lines 68 - 76 and 76 - 90 there is some lack of low in the writing that makes it difficult to follow. I acknowledge that this is in part from adding in new information in response to reviewer requests but feel the narrative could flow more smoothly with some minor tweaks to the structuring of these paragraphs. The paper is important and well written and these early paragraphs seem to lack the same strength.

Reviewer #3: (No Response)

7. PLOS authors have the option to publish the peer review history of their article (what does this mean? ). If published, this will include your full peer review and any attached files.

**Do you want your identity to be public for this peer review?** For information about this choice, including consent withdrawal, please see our Privacy Policy .

Reviewer #1: No

Reviewer #2: No

Reviewer #3: No

---

## [Editor Report · Acceptance letter]

PONE-D-25-35455R1

PLOS ONE

Dear Dr. Rodas,

I'm pleased to inform you that your manuscript has been deemed suitable for publication in PLOS ONE. Congratulations! Your manuscript is now being handed over to our production team.

Kind regards,

on behalf of

Dr. Rosemary Bassey

Academic Editor

PLOS ONE